# Efficacy of Catheter Ablation Using the Electroanatomical System without the Use of Fluoroscopy in Patients with Ventricular Extrasystolic Beats

**DOI:** 10.3390/jcm12144851

**Published:** 2023-07-24

**Authors:** Dariusz Rodkiewicz, Edward Koźluk, Karol Momot, Karolina Rogala, Agnieszka Piątkowska, Małgorzata Buksińska-Lisik, Przemysław Kwasiborski, Artur Mamcarz

**Affiliations:** 1Department of Cardiology and Internal Diseases, Regional Hospital in Miedzylesie, 04-749 Warsaw, Poland; 2Chair and Department of Experimental and Clinical Physiology, Laboratory of Centre for Preclinical Research, Medical University of Warsaw, 02-097 Warsaw, Poland; 3Department of Emergency Medicine, Wroclaw Medical University, 50-556 Wroclaw, Poland; 43rd Department of Internal Medicine and Cardiology, Medical University of Warsaw, 04-749 Warsaw, Poland

**Keywords:** catheter ablation, fluoroscopy elimination, electroanatomic mapping, ventricular extrasystolic beats

## Abstract

Background: Catheter ablation (CA) has become safe and efficient for the treatment of patients with ventricular extrasystolic beats (VEBs). The three-dimensional electroanatomic mapping (EAM) system allows the elimination of fluoroscopy time during CA procedures. Non-fluoroscopy CA is a challenging procedure requiring intimate knowledge of cardiac anatomy in patients with VEBs. The study aimed to evaluate the efficacy and safety of the non-fluoroscopy CA using the EAM system in patients with VEBs. Methods: Completely fluoroless CA of VEBs guided by EAM was performed in 86% (94 out of 109) of consecutive patients with VEBs. The remaining 15 patients underwent conventional fluoroscopy-guided CA. Demographic and clinical baseline characteristics, procedure parameters, and following complications were obtained from the medical records. Primary outcomes were the acute procedural success rate, the permanent success rate (6-month follow-up), complications, and procedure time. Results: There were no significant differences between groups regarding baseline characteristics. Acute procedural success was achieved in 85 patients (90%) in the non-fluoroscopy group and in 14 patients (93%) in the fluoroscopy group (ns). A long-term success rate was achieved in 82 patients (87%) in the non-fluoroscopy group and in 14 (82%) patients in the fluoroscopy group (ns). The median procedure time was 85 min in the non-fluoroscopy group and 120 min in the fluoroscopy group (*p* = 0.029). There was only one major complication in the non-fluoroscopy group (ns). Conclusions: Completely fluoroless CA of VEBs guided by EAM is a feasible, safe, and efficient procedure.

## 1. Introduction

Catheter ablation (CA) has become the established therapy for patients with symptomatic ventricular extrasystolic beats (VEBs) [1]. CA of idiopathic VEBs originating from different sites within the left ventricle (LV) or right ventricle (RV) requires intimate knowledge of cardiac anatomy [2].

The EAM system is helpful to create precise anatomy of the heart chamber. Additionally, EAM allows the creation of an activation map of the arrhythmia origin, navigates catheters precisely, and reduces or even eliminates radiation exposure [3]. However, additional fluoroscopy navigation could be helpful to check or save the position of the catheter in difficult cases, especially in organic heart disease with atypical anatomy of the ventricle. Unfortunately, it is related to ionizing radiation.

There are differences in the effectiveness of CA depending on the arrhythmia location [4]. Because of the thicker myocardium of the LV as compared to the RV, the effectiveness of CA is lower in the LV. Furthermore, LV arrhythmia can be located in either endocardial or epicardial sites. If the arrhythmia origin is epicardial, precise location and ablation of the arrhythmia site can be challenging with the use of an endocardial approach [5]. The most common approach to epicardial arrhythmia ablation is through the coronary sinus [6]. However, it is associated with difficulties, because of the anatomical proximity of the coronary arteries and the presence of a pericardial fat pad that can limit radiofrequency (RF) energy delivery. The success rate of CA is also lower when the arrhythmia origin is near the atrioventricular node or bundle branch. In these locations, the conductive system is more likely to be destroyed during RF ablation compared to cryoablation [7]. It is because cryomapping is reversible and useful in vulnerable arrhythmia locations, especially near the conductive system or coronary arteries. EAM allows the performing of both RF CA and cryoablation of VEBs. If VEBs originate from the right ventricular outflow tract (RVOT), the effectiveness of CA exceeds 80% [8]. Although the CA of VEBs is safe, there is a possibility of minor or major complications. Minor complications include pericardial effusion, vascular access complications (groin hematoma, pseudoaneurysm, arterio-venous fistula), right bundle branch block (RBBB), and left bundle branch block (LBBB). Major complications include pericardium tamponade, atrioventricular (AV) block, major bleeding, thromboembolic event, myocardial infarction, and aorta dissection [9]. 

There are limited data about the safety and efficacy of CA without fluoroscopy guided by the EAM system in patients with VEBs from various arrhythmia locations [3,10].

The present study aimed to assess the feasibility, safety, and efficacy of completely fluoroless CA of VEBs guided by the EAM system in adult patients.

## 2. Methods

### 2.1. Study Design and Study Population

The study included 109 patients with symptomatic VEBs, who were qualified for the ablation procedure based on 24 h ECG Holter (more than 10.000 VEBs per day). The purpose of the study was to perform every procedure following the ALARA rule of using fluoroscopy (as low as reasonably achievable). Only when there was a necessity arising from the factors described below, fluoroscopy was used. Completely fluoroless CA of VEBs guided by the EAM system was performed in 94 (86%) consecutive patients with VEBs, whereas conventional fluoroscopy-guided CA was performed in 15 (13,8%) such patients. Procedures were performed between January 2020 and September 2022. Participants signed written informed consent forms. Demographic and clinical data were obtained from the medical records.

### 2.2. Ablation Procedure

In all patients qualified for CA, antiarrhythmic drugs were discontinued for a period of at least 5 half-lives before hospital admission. 

For patients with left-sided VEBs, retrograde aortic access was achieved by femoral artery puncture. Then, a vascular sheath of 7F or 8F, depending on the size of the ablation electrode, was introduced and a fast anatomical map of the aorta using the ablation electrode was performed, especially in order to locate the aortic valve and ostium coronary arteries and, thus, enhance the safety of the procedure. In case of difficulties in passing through the aorta and/or the aortic valve, fluoroscopy for guidance was used. Then, using the ablation catheter, an electroanatomic map of the earliest activation site of the arrhythmia origin was created. Intravenous unfractionated heparin in a bolus dose of 5000 U and infusion rate based on the activated clotting time (ACT) was administered.

In right-sided VEBs, access to the RV was obtained through a femoral vein access without heparin during the procedure. A vascular sheath of 7F or 8F, depending on the type of ablation catheter, was introduced into the right femoral vein. Initially, the ablation catheter was inserted into the femoral vein and then advanced into the inferior vena cava until reaching the right atrium. In some cases, a fast anatomical map of the right atrium (RA) was used to facilitate navigation in case the catheter dislodged from the RV into the RA. In such instances, a return pathway was determined. Introducing the catheter into the RVOT location sometimes required special maneuvering due to anatomical difficulties. In most cases, while mapping the RA, the tricuspid valve and the bundle of His were marked, and then, depending on the arrhythmia’s location, the RV or RVOT was mapped. The electroanatomical map was created to find the site of the earliest activation of the ventricular arrhythmia’s origin, which was marked with a color point in the EAM.

RF procedures were conducted with the CARTO system 3 System (Biosense Webster, Diamond Bar, California, USA) or the EnSite NavX system (Abbott Medical, Minneapolis, MN, USA).

The Smartablate RF generator and irrigated ablation catheters (Smarttouch, Navistar, Thermocool) were used during the CARTO CA. Power outputs ranged from 30 to 45 W with a temperature not exceeding 48 °C. Using the ablation catheter, we created an anatomical map of the ventricle and collected points during the VEBs, which revealed the site of the earliest activation of VEBs. We marked this focal arrhythmia with red color surrounded radially by yellow and green. This location served as the target for the RF application.

During the EnSite NavX CA, an Ampere RF generator and both irrigated and non-irrigated ablation catheters (4 mm ablation catheters: Hagmed, Rawa Mazowiecka, Poland or FlexAbility, Abbott Medical, Minneapolis, MN, USA) were used. The EnSite NavX system, in contrast to the CARTO system, allowed for visualization of the catheter from its introduction into the vascular sheath. In this way, the pathway of the catheter was tracked from the beginning of the procedure. In the EnSite NavX system, in contrast to the CARTO mapping system, the site of the earliest activation was marked in white. It is also possible to perform cryoablation in this system.

Cryoablations were performed using the EnSite NavX system with 7 French Freezor catheters (Medtronic, Inc., Minneapolis, MN, USA). The ablations were preceded by a cryomapping procedure at −30 °C to control the efficacy and safety. The temperatures during cryoablation reached less than −70 °C to obtain a permanent ablation effect.

If needed, conventional fluoroscopic mapping was performed with an X-ray system utilizing the customized settings for each patient to achieve minimal radiation dose compatible with appropriate image quality. 

Arrhythmia characteristics, procedure time, ablation time, number of ablation applications, fluoroscopy time, dose area product, procedural success rates, and complications were analyzed in the study. Procedure time was measured from the time of local anesthesia to the removal of vascular sheaths (skin to skin). VEBs location was classified as right ventricular outflow tract (RVOT), right ventricle (RV), aortomitral continuity (A-M), aorta (Ao), left ventricle (LV), para-hisian, and epicardial.

Acute procedural success was defined as a clinically significant reduction in targeted VEBs within 30 min after the last ablation application. All patients underwent a post-procedural echocardiogram. Minor and major complications that occurred up to the hospital release were included in the results of the study. Then, 24 h ECG Holters were performed after 2 and 6 months after procedures. Long-term success was defined as the reduction in targeted VEBs below 1.000 in 24 h ECG Holter after a 6-month follow-up. Additionally, in some patients, a 12-month follow-up was reached.

### 2.3. Statistical Analysis

Statistical analysis was performed using Statistica software version 13.3. Quantitative variables are presented as means and standard deviations (SD) or standard errors (SE), as well as medians (interquartile range—IQR). The ANOVA test (normal distribution) and the Kruskal–Wallis H test (non-normal distribution) were performed in the comparison of variables between the groups. Categorical variables are expressed as numbers and percentages and compared using the χ^2^ test. The Shapiro–Wilk test was used to assess the normality of data. The equality of variances was assessed by the Levene test. The differences were considered statistically significant if *p* < 0.05.

## 3. Results

There were no statistical differences between the groups regarding demographic and clinical characteristics (Table 1). 

The characteristics of the procedure are presented in Table 2. Acute procedural success was achieved in 90% (85 out of 94) of patients from the non-fluoroscopy group and 93% (14 out of 15) of patients from the fluoroscopy group (ns). Long-term procedural success was achieved in 87% (82 out of 94) of patients from the non-fluoroscopy group and in 93% (14 out of 15) of patients from the fluoroscopy group (ns). The median procedure time (IQR) was 85 (65–119) minutes in the non-fluoroscopy group and 120 (75–175) minutes in the fluoroscopy group (*p* = 0.029). 

The arrhythmia characteristics are presented in Table 3. RVOT VEBs were discovered in 46% (50 out of 109) of patients. In 96% (48 out of 50) of patients, CA was performed without fluoroscopy. In the non-RVOT location, CA without fluoroscopy was performed in 78% (46 out of 59) of patients (*p* = 0.006).

An amount of 12 para-hisian locations and 16 epicardial location cases were revealed. Due to catheter instability and inducing nodal rhythm, fluoroscopy was required in three out of twelve para-hisian ablation locations. Due to the need for coronary angiography, left-sided locations, or catheter instability, fluoroscopy was used in three out of sixteen epicardial locations. Long-term success was achieved in 100% (12 out of 12) of para-hisian applications and in 44% (7 out of 16) epicardial applications. In the non-fluoroscopy ablations of epicardial locations, long-term success was achieved in five out of thirteen cases (38%), whereas in the fluoroscopy-guided ablations, long-term success was achieved in two out of three cases (67%). In the case of left-sided VEBs, A-M locations were found in 12 cases. In this group, it was possible to perform 11 procedures without the use of fluoroscopy, and the long-term success rate was achieved in nine out of eleven cases (82%). In the fluoroscopy-guided ablations, one procedure was also successful. 

Only one major complication occurred in the non-fluoroscopy group and none in the fluoroscopy group (*p* = 0.483). This one major complication was cardiac tamponade after RF CA with the power of 40 W in LV, which was successfully punctured. Cardiac surgery was unnecessary.

Figure 1 represents the subdomain Kaplan–Meier 12-month survival analysis. The 12-month period of event-free survival was assessed in sixty-two patients from the non-fluoroscopy group and seven patients from the fluoroscopy group.

## 4. Discussion

The majority of the papers about non-fluoroscopy ablations focus on supraventricular arrhythmias and refer to single-center observations [11,12].

The present study described the procedures and outcomes of performing CA in a wide variety of ventricular arrhythmias without the use of fluoroscopy. It was observed that the RVOT location is favorable for performing the procedure without the use of fluoroscopy. Experience showed that most procedures can be performed without fluoroscopy and that the only limitations are the operator’s skill and mentality. A thorough understanding and knowledge of the heart anatomy is crucial [4].

This approach should be included in all electrophysiology training programs to reduce exposure to radiation and decrease the risk of late radiation complications, including cancer.

In this study, cardiologists focused on the ALARA rule of using fluoroscopy [13]. Ionizing radiation affects human cells in two ways: directly and indirectly. Direct exposure to ionizing radiation causes changes in the deoxyribonucleic acid (DNA) and changes in the structure of nucleotides. Indirect exposure induces the formation of free oxygen radicals that can cause changes in the next few nucleotides, contributing to damage in many places. Ionizing radiation also impairs the efficiency of repair mechanisms, leading to apoptosis or carcinogenesis [14]. 

The effect of ionizing radiation on organisms depends on the type of tissue. Young people, patients with diseases of the hematopoietic system, patients with cancer, and pregnant women are in groups with high risks of complications of ionizing radiation. There is evidence that even small doses have a harmful effect on pregnant women, especially in the early embryonic period and during organogenesis [15]. It is worth noting that in our research group, there were 39 women of reproductive age (up to 49 years old) who may not have been aware of their pregnancy. Exposure to ionizing radiation during organogenesis can lead to serious birth defects, including distortion of organs and impairment of their functions, which is why we believe in striving to eliminate fluoroscopy in such cases.

In our study, the EAM system was used in patients to reduce ionizing radiation. Non-fluoroscopy ablation is a safe and effective procedure and should be recommended especially in young patients and pregnant women [16]. Non-fluoroscopy ablation should become a standard procedure for patients with VEBs from both sides of the heart.

As a result, 86% (94 out of 109) of our patients were treated without radiation. Procedures without fluoroscopy were significantly shorter than those with fluoroscopy. The reason for this could be that patients who underwent the procedure without fluoroscopy may have had less complex anatomical conditions compared to those who required conversion to a procedure with fluoroscopy. Consequently, less time was required for precise mapping and ablating the origin site of the arrhythmia. In case of anatomical difficulties, the operator was forced to use fluoroscopy. Because of the atypical anatomy of the heart, epicardial arrhythmia locations, and the catheter instability during application lower efficacy of CA may be observed and, often, fluoroscopy needs to be used. However, for experienced operators, these differences may be insignificant. It was found that fluoroscopy did not improve the efficacy of the CA. In our view, fluoroscopy allows performing safe ablation in single cases connected with difficult anatomical conditions, epicardial arrhythmia origin, and catheter instability. Based on our results, we suggest that the epicardial location is associated with lower procedure success rates. Probably, the use of fluoroscopy in these locations could improve effectiveness. However, further studies on a larger patient group with this type of arrhythmia are needed.

Among 109 procedures, cardiologists were forced to use fluoroscopy in 15 cases. In spite of this, the use of fluoroscopy was minimal, with an average time of 2.18 ± 0.62 min. The maximum time recorded was 9.67 min.

The use of fluoroscopy was necessary in case of difficulties in navigating the catheter through the arteries. This was caused by scattered atherosclerotic lesions in the iliac, femoral, and aortic vessels, especially in the sites of bifurcations. When resistance was discovered during catheter passage, fluoroscopy was used. Additionally, an alternative solution to overcome this issue could be the use of a hydrophilic guidewire and a long vascular sheath.

Another challenge may arise when passing through the aortic valve, and in cases where the operator detected resistance near that location. Fluoroscopy was used to ensure a safe passage through this structure, paying special attention to the coronary arteries. A method that often allows for resolving this issue is bending the ablation catheter in the aortic arch and entering the valve with “a loop”.

Another problem is the presence of numerous venous collaterals, causing the catheter to deviate from the correct path and enter side branches. The use of fluoroscopy allows us to avoid damaging the vein.

Another scenario where fluoroscopy is used is when there is a short distance to the coronary artery, requiring coronarography. RF ablation is not recommended if the arrhythmia location is within 1 cm of the course of a coronary artery. Application may lead to a spasm, thrombosis, or mechanical vessel wall damage. An alternative approach to address this issue is the use of focal cryoablation. Additionally, the use of coronarography is necessary in cases of epicardial arrhythmia location and mapping within the venous system of the heart. In the case of epicardial RF ablation, it is essential to exclude the proximity of coronary arteries’ course.

The location of arrhythmias in the papillary muscle could be challenging due to catheter instability. In such cases, using a long stabilizing sheath can be a proper solution.

An experienced cardiologist is capable of performing the procedure without the use of fluoroscopy in both the right and left ventricles.

In light of the prevailing consensus that there is no threshold for safe radiation dose, efforts should be made to eliminate the use of fluoroscopy not only from complex ablation procedures but also in simpler arrhythmic substrates, such as VEBs, where the use of EAM has been considered. The study’s promising results inspire us to further investigate the utility of EAM mapping in different clinical conditions. 

## 5. Conclusions

The non-fluoroscopy approach is feasible, safe, and effective by integrating EAM with the operator’s mentality, skill, and knowledge of intracardiac anatomy.

The RVOT location is favorable for performing non-fluoroscopy ablation. Additionally, the use of fluoroscopy can be avoided in different and more challenging arrhythmia locations even requiring access to the left heart chambers. 

## Figures and Tables

**Figure 1 jcm-12-04851-f001:**
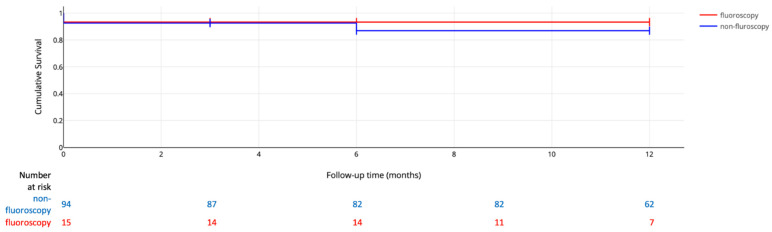
Kaplan–Meier Survival Analysis.

**Table 1 jcm-12-04851-t001:** Baseline patient characteristics and clinical assessment.

	Non-Fluoroscopy Group (n = 94)	Fluoroscopy Group (n = 15)	*p* Value
Sex (Male) n (%)	33 (35)	9 (60)	0.066
Age (years) Mean (SE)	51 (1.55)	56 (2.66)	0.253
Left Ventricle Ejection Fraction Mean (%), (SE)	53 (0.78)	49 (2.64)	0.054
Hypertension n (%)	36 (38)	6 (40)	0.589
Obesity n (%)	8 (9)	2 (13)	0.548
Lipidemic disorders n (%)	30 (32)	5 (33)	0.913
Diabetes Mellitus type 2 n (%)	9 (10)	2 (13)	0.654
Coronary artery Disease n (%)	15 (16)	5 (33)	0.106
Thyroid disease n (%)	12 (13)	0 (0)	0.142

Abbreviations: SE—Standard error.

**Table 2 jcm-12-04851-t002:** Procedure characteristics.

	Non-Fluoroscopy Group (n = 94)	Fluoroscopy Group (n = 15)	*p* Value
Procedure time, (minutes, median, IQR)	85 (65–119)	120 (75–175)	0.029
Application time (minutes, median, IQR)	6.15 (3–14)	5.85 (2–22)	0.393
Number of applications, (median, IQR)	6 (2–11)	7 (1–19)	0.491
EAM CARTO n (%)	84 (89)	15 (100)	0.185
EAM EnSite NavX n (%)	10 (11)	0	0.185
Cryoablation n (%)	5 (5)	0	0.360
RF ablation n (%)	89 (95)	15 (100)	0.360
RF ≥ 40 W n (%)	32 (34)	1 (7)	0.032
Acute procedural success n (%)	85 (90)	14 (93)	0.717
Long-term success n (%)	82 (87)	14 (93)	0.743
Major complication n (%)	1 (1)	0	0.483

Abbreviations: IQR—interquartile range, EAM—Electroanatomic mapping RF—radiofrequency.

**Table 3 jcm-12-04851-t003:** Arrhythmia characteristics.

	Non-Fluoroscopy Group (n = 94)	Fluoroscopy Group (n = 15)	*p* Value
RVOT location n (%)	48 (51)	2 (13)	0.023
RV (except RVOT) n (%)	8 (9)	4 (27)	
Ao n (%)	4 (4)	2 (13)	
A-M n (%)	11 (12)	1 (7)	
LV (except M-A) n (%)	23 (24)	6 (40)	
Precise location:			
Para-hisian location n (%)	9 (10)	3 (20)	0.231
Epicardial location n (%)	13 (14)	3 (20)	0.531

Abbreviations: RVOT—Right ventricular outflow tract, RV—right ventricle, Ao—Aorta, A-M—Aortomitral continuity, LV—left ventricle.

## Data Availability

The data that support the findings of this study are available from the corresponding author [K.M].

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
