# Peer review of "Efficacy of Catheter Ablation Using the Electroanatomical System without the Use of Fluoroscopy in Patients with Ventricular Extrasystolic Beats"

_jcm, 2023, doi:10.3390/jcm12144851_

Round 1

Reviewer 1 Report

First, I would like to thank the Editor for offering me the opportunity to review this interesting paper.

I want to congratulate to the Authors for their work. This is crucial topic in current EP, and any effort in promoting radiation awareness are welcome.

However, some issues need to be addressed.

I would suggest to broaden the Discussion section. Please focus on the danger of radiation use of both the patients and the EP lab personnel and on the strategies, along with use of electroanatomic mapping systems, to reduce radiation exposure during EP procedures.

I have some doubts on the need of the comparison between the procedures with and without the use of fluoroscopic guidance. This is a nonrandomized comparison, and, as clearly stated by the Authors in the Discussion section, the difference in the use of fluoroscopy was most probably driven by differences in the complexity of the procedure and anatomical origin of the arrhythmia. Moreover, no correction of the baseline characteristics was performed during the statistical analysis. The main message of this study is that electroanatomic mapping system can be successfully implemented for ablation of ventricular premature contraction and are a useful tool for the reduction of X-rays use in the EP lab. I believe that the aforementioned comparison is not helpful and does not add any further element to the study.

In Table 3 what does the p-value of 0.023 refer to? I cannot understand what the Author compared there?

Minor comments

In all the tables and throughout the manuscript please explain all the acronyms. For example, “ns” = non significant.

In the “Results” section please report the p-values instead of

Line 85: please modify “retrograde aortal access” in “retrograde aortic access”.

Line 94: please modify “was used femoral vein access to achieve the RV” in “access to the RV was obtained through a femoral vein access”.

In general, I suggest checking and revising the English language throughout the manuscript.

English Language in the manuscript need moderate editing.

Author Response

Dear Reviewer

Thank you for your valuable suggestions and for recognizing the significance of reducing fluoroscopy in ablation procedures. We have taken your suggestion into account and have expanded the discussion section of our manuscript:

  1.     We have focused on the risks associated with radiation exposure for both patients and EP lab personnel during ablation procedures.
  2.     We have also discussed additional suggestions and tips for avoiding the use of fluoroscopy.
  3.     We have made the language changes to the text.
  4.     We have added P values to all results.
  5.     We have added the list of all abbreviations at the end of the manuscript.

The value of P (0.023) was calculated using the chi-square test of independence. This test was chosen because the variables were categorical (nominal). In this case, the degree of freedom was 4, and the chi-square statistic was 11.2953. The test calculates the overall significance between the two groups (the non-fluoroscopy and the fluoroscopy), determining whether there is a statistically significant difference in locations in general. The answer is "yes" because the p-value is less than 0.05. Post-hoc tests for the chi-square independence test are not widely used. The P-value in this comparison shows that there are differences between arrhythmia locations between the two groups.

While we understand that this comparison between the two groups may not be the primary focus of our study, we believe it still adds value to the knowledge associated with performing non-fluoroscopy procedures:

  1.   The comparison was made for reference only and it showed that comorbidities and patient characteristics do not have a statistically significant impact on whether the procedure will require fluoroscopy. Even features such as LVEF or age were not different in both groups.
  2.     We have shown that the use of fluoroscopy can be avoided in almost every location (Table 2). This is a fairly novel perspective, especially for left-sided VEBs. Some operators believe that all left-sided VEBs ablation procedures should be performed with the use of fluoroscopy. It is considered necessary due to the safety of crossing the aortic valve with the catheter. However, proper looping of the catheter in the aortic arch allows for safe passage through that area without the risk of entrapment. Our comparison showed that non-fluoroscopy procedures can be performed on both sides of the heart.
  3.   There is no surprise that the procedure time in the second group was longer. However, the comparison shows that time and number of applications were similar. This confirms that fluoroscopy is used only for maneuvering the ablation catheter to find the best application spot and not for increasing the effectiveness of the ablation.
  4.   In response to your doubts, we have included a detailed description of the difficulties in individual patients that made us use fluoroscopy. We aimed to offer a more comprehensive understanding of the factors leading to the use of fluoroscopy during the procedures.

Thank you once again for recognizing the significance of radiation exposure in ablation procedures, and we hope that our response and changes in the manuscript will be satisfactory for you.

Reviewer 2 Report

This is a study of 109 patients with frequent PVCs with attempted fluoroless ablation, in 15 of them fluoroscopy being used mainly for safety reasons.

Major issues:

- irrelevant comparison between the 2 groups, the so called fluoroscopy group was actually represented by the patients with more difficult anatomy and/or location of the arrhythmic focus, therefore is no surprise that the procedure times were longer

- unnecessary/irrelevant auto-citations

Author Response

Dear reviewer

Thank you very much for taking the time to review our article. We sincerely appreciate it. thank you for paying attention to auto-citations. However, the second reviewer commended the idea and emphasized that any work describing a reduction in radiation burden is highly desirable. We understand that self-citations of studies that address similar topics in this case can increase interest in the subject of non-fluoroscopy procedures and it will potentially change the approach to this topic of many EP operators.

While we understand that this comparison between the two groups may not be the primary focus of our study, we believe it still adds value to the knowledge associated with performing non-fluoroscopy procedures:

  1.     The comparison was made for reference only and it showed that comorbidities and patient characteristics do not have a statistically significant impact on whether the procedure will require fluoroscopy. Even features such as reduced LVEF or older age were not different in both groups.
  2.     We have shown that the use of fluoroscopy can be avoided in almost every location (table 2). This is a fairly novel perspective, especially for left-sided VEBs. Some operators believe that all left-sided VEBs ablation procedures should be performed with the use of fluoroscopy. It is considered necessary due to the safety of crossing the aortic valve with the catheter. However, proper looping of the catheter in the aortic arch allows for safe passage through that area without the risk of entrapment. Our comparison showed that non-fluoroscopy procedures can be performed on both sides of the heart.
  3.   There is no surprise that the procedure time in the second group was longer.  However, the comparison shows that time and number of applications were similar. This confirms that fluoroscopy is used only for maneuvering the ablation catheter to find the best application spot and not for increasing the effectiveness of the application.
  4.   In response to your doubts, we have included a detailed description of the difficulties in individual patients that made us use fluoroscopy. We aimed to offer a more comprehensive understanding of the factors leading to the use of fluoroscopy during the procedures.

Thank you once again for taking the time to review our article, and we sincerely hope that our response will be satisfactory to you.

Round 2

Reviewer 2 Report

-Unnecessary and/or inappropriate self-citation stil present (see reference 3, 5).

- Excessive discussion on effects of ionising radiation

- CARTO system also allows catheter tracking from the pelvis level (line 271)

Author Response

Dear Reviewer,

Thank you for your feedback. We have addressed the issues as follows:

  • Unnecessary and/or inappropriate self-citation (References 3 and 5): 
    • We have removed these citations as suggested.
  • Excessive discussion on ionizing radiation effects: 
    • We have significantly reduced this section.
  • CARTO system catheter tracking from the pelvis level (line 271): 
    • You are absolutely correct, the ability to track the catheter from the pelvis level in the CARTO system is possible under specific electromagnetic field orientation. However, we agree with your concern, and to avoid any confusion, we have removed that specific fragment from our manuscript.

Thank you for your valuable input.

Best regards,
